# Exploring Heterogeneity of Fecal Microbiome in Long COVID Patients at 3 to 6 Months After Infection

**DOI:** 10.3390/ijms26041781

**Published:** 2025-02-19

**Authors:** Jelle M. Blankestijn, Nadia Baalbaki, Rosanne J. H. C. G. Beijers, Merel E. B. Cornelissen, W. Joost Wiersinga, Mahmoud I. Abdel-Aziz, Anke H. Maitland-van der Zee

**Affiliations:** 1Department of Pulmonary Medicine, Amsterdam UMC, University of Amsterdam, 1105 AZ Amsterdam, The Netherlands; j.m.blankestijn@amsterdamumc.nl (J.M.B.);; 2Amsterdam Institute for Infection and Immunity, 1105 AZ Amsterdam, The Netherlands; 3Amsterdam Public Health, 1081 HV Amsterdam, The Netherlands; 4Department of Respiratory Medicine, NUTRIM Institute of Nutrition and Translational Research in Metabolism, Maastricht University Medical Centre+, 6229 ER Maastricht, The Netherlands; 5Center for Experimental and Molecular Medicine, Amsterdam UMC, University of Amsterdam, 1105 AZ Amsterdam, The Netherlands; 6Department of Internal Medicine, Division of Infectious Diseases, Amsterdam UMC, University of Amsterdam, 1105 AZ Amsterdam, The Netherlands; 7Department of Clinical Pharmacy, Faculty of Pharmacy, Assiut University, Assiut 71526, Egypt; 8Department of Pediatric Respiratory Medicine, Emma Children’s Hospital, Amsterdam UMC, University of Amsterdam, 1105 AZ Amsterdam, The Netherlands

**Keywords:** microbiome, long COVID, heterogeneity, butyrate-producing bacteria, butyrate, lung function

## Abstract

An estimated 10% of COVID-19 survivors have been reported to suffer from complaints after at least three months. The intestinal microbiome has been shown to impact long COVID through the gut–lung axis and impact the severity. We aimed to investigate the relationship between the gut microbiome and clinical characteristics, exploring microbiome heterogeneity through clustering. Seventy-nine patients with long COVID evaluated at 3 to 6 months after infection were sampled for fecal metagenome analysis. Patients were divided into two distinct hierarchical clusters, based solely on the microbiome composition. Compared to cluster 1 (n = 67), patients in cluster 2 (n = 12) showed a significantly reduced lung function (FEV_1_, FVC, and DLCO) and during acute COVID-19 showed a longer duration of hospital admissions (48 compared to 7 days) and higher rates of ICU admissions (92% compared to 22%). Additionally, the microbiome composition showed a reduced alpha diversity and lower proportion of butyrate-producing bacteria in cluster 2 together with higher abundances of *Ruminococcus gnavus*, *Escherichia coli*, *Veillonella* spp. and *Streptococcus* spp. and reduced abundances of *Faecalibacterium prausnitzii* and *Eubacteria* spp. Further research could explore the effect of pre- and pro-biotic supplementation and its impact on lung function and societal participation in long COVID.

## 1. Introduction

According to the World Health Organization, there have been over 776 million confirmed cases of infection with the severe, acute respiratory syndrome of coronavirus 2 (SARS-CoV-2) as of September 2024. The disease resulting from this infection is also known as coronavirus disease 19 (COVID-19) [1]. Due to decreased and limited testing in asymptomatic cases, the number of infections might be an underestimation. It is estimated that approximately 10–45% of all COVID-19 survivors continue to suffer from complaints at least 3 months after infection, resulting in a condition known as post-acute sequelae of COVID-19, post-COVID-19 condition or long COVID [2,3]. The manifestation of long COVID is highly heterogeneous, and patients suffer from complaints related to various organ systems, including fatigue, cardiovascular, respiratory and neurological symptoms [4,5].

One of the factors related to the severity of both COVID-19 and long COVID is the gastrointestinal microbiome [6,7]. The gastrointestinal microbiome has been shown to influence respiratory diseases such as asthma, chronic obstructive pulmonary disease (COPD), pneumonia and lung cancer, in both harmful and beneficial manners [8,9]. This is through the so-called gut–lung axis mediated by small metabolites known as short-chain fatty acids (SCFAs) [9,10], including butyrate, propionate and acetate. The microbiome is highly versatile and is easily influenced by factors such as culture, diet, disease and antibiotic use. Several studies have already identified particular bacterial species, such as *Ruminococcus gnavus*, *Faecalibacterium prausnitzii* and the *Veillonella* genus, that are related to the severity of long COVID [7,11], highlighting the role of the microbiome in the pathophysiology of the disease.

Due to the heterogeneity in long COVID, a one-size-fits-all treatment for long COVID likely will not be developed. Unsupervised clustering of patients based on their clinical characteristics, molecular profiles or microbial composition could help to identify patients with a similar disease pathophysiology, which could contribute to the development of the optimal treatment strategy tailored to the individual patient in a precision medicine approach. In this study, we aimed to investigate the relationship between the gastrointestinal microbiome of patients with long COVID at 3 to 6 months after the initial SARS-CoV-2 infection and their clinical characteristics. Additionally, we aimed to perform unbiased, unsupervised clustering in order to explore the heterogeneity within the gut microbiome of long COVID.

## 2. Results

Of the 95 patients from the P4O2 COVID-19 study, 79 were able to provide a fecal sample with sufficient quality for analysis. The patient characteristics of this population can be found in Table 1. Our patient population had an even distribution in terms of sex, an average age of 54.6 (±6.0) years and an average body mass index (BMI) of 30.4 (±5.2). These patients most commonly reported respiratory complaints (84%), fatigue (72%) and neurological complaints (70%). Nearly all patients were admitted to the hospital during acute COVID-19 (91.1%). In terms of microbiome composition, the phylum *Firmicutes* is the most common, comprising nearly two thirds (66%) of taxa found, followed by *Actinobacteria* (17%) and *Bacteroidetes* (13%). On a genus level, there is less of a domination of a single genus: nearly half (46%) of the microbiome composition is made up of genera with a relative abundance less than 2%, while the most common genera are *Faecalibacterium* (9%) and *Bifidobacterium* (8%) (Figure 1).

### 2.1. Microbiome Composition Showed Associations with Lung Function and Severity of Acute COVID-19

The general microbiome composition in our patient population was compared using the alpha and beta diversity. The microbiome composition primarily exhibited differences in relation to lung function and the severity of acute COVID-19. A significantly lower Shannon index was observed in patients with an abnormal FVC or DLCO and a severe classification during the acute COVID-19 phase. These variables also showed a significantly lower richness with antibiotic use and an abnormality in at least one pulmonary function test measure also showing a significantly lower diversity (Figure 2A,C). This was further confirmed with the beta diversity, where all these variables demonstrated significant separation based on the Bray–Curtis distance (Figure 2B,D).

### 2.2. Unsupervised Clustering Showed Differences Similar to Alpha and Beta Diversity

Based on the fecal microbiome profiles, the patient population was divided into two clusters. A distinction in two clusters showed a higher silhouette index and a higher cluster stability compared to adding more clusters (except for 3 clusters, Appendix A). These clusters were imbalanced in terms of size, where the majority was placed in cluster 1 (67 patients), with 12 patients in cluster 2 (Appendix A). The patient characteristics for each cluster are found in Table 2. Cluster 2 contained slightly more males compared to cluster 1; however, this was not statistically significant (*p* = 0.057). There were no differences in terms of age, BMI or time between infection and follow-up. In accordance with the analyses into the alpha and beta diversity, the main differences in the cluster were seen in the lung function and the acute phase of COVID-19, where a significantly lower lung function for the FEV_1_, FVC and DLCO (Figure 3) and more severe phenotype during acute COVID-19 were observed in cluster 2. They had a longer hospital stay (48 days compared to 7 days), were admitted to the ICU more often (92% compared to 22%), showed a higher rate of complications and were administered antibiotics more often during their infection than in cluster 1 (75% compared to 25%). At the 3 to 6 months follow-up, patients in cluster 2 also showed more signs of bronchiectasis on a CT-scan (58% compared to 16%). Although not significant, patients in cluster 2 showed signs of a decreased quality of life (EQ-5D, *p* = 0.084) and participation (USER-P, *p* = 0.114), but in contrast, they showed less anxiety (*p* = 0.092). When examining only those patients admitted to the ICU, we found that most of the conclusions regarding the entire cohort also apply to this subset of patients or are even more significant (Appendix A).

### 2.3. Fecal Microbiome of Cluster 2 Exhibited Lower Diversity and Contained Less Butyrate-Producing Bacteria

Next, we analyzed differences in the microbiome composition between the clusters. The diversity of the fecal microbiome from patients in cluster 2 was significantly lower than in cluster 1 (*p* < 0.001) and showed significant separation in terms of beta diversity (*p* < 0.001, Figure 4A,B). Differential abundance analysis using ANCOM-BC revealed 265 differentially abundant taxa, the majority of which (213 taxa) were more abundant in cluster 1. A full list of differentially abundant taxa can be found in the Appendix A. In cluster 1, we found higher abundances of species from the *Coprococcus* and *Eubacterium* genera, as well as *Faecalibacteria*, including the *prausnitzii* and *intestinalis* species. In cluster 2, we found increased abundances of *Ruminococcus gnavus*, *Veillonella*, *Streptococcus* and *Escherichia coli.* Upon further inspection of the differentially abundant taxa, we noticed an enrichment of butyrate-producing species present within the taxa that are more abundant in cluster 1 (Figure 4C). While analyzing the proportion of butyrate-producing taxa per patient, we found that patients in cluster 1 did indeed exhibit a larger proportion of butyrate-producing taxa than patients in cluster 2 (*p* = 0.011, Figure 4D). When only studying the patients admitted to the ICU, we still observed a significantly lower proportion of butyrate-producing bacteria in cluster 2 (*p* = 0.047, Appendix A).

## 3. Discussion

In this study, we explored the microbiome composition of patients with long COVID. We found that patients with at least one abnormality in the lung function test measures or a severe, acute COVID-19 showed a decreased alpha diversity and a separation for the beta diversity compared to the other patients without these abnormalities. Unbiased clustering revealed two clusters that displayed these same trends. The cluster with a decreased lung function and severe acute COVID-19 also showed slightly more respiratory symptoms, a marginally decreased quality of life and societal participation but fewer signs of anxiety. This cluster also showed a lower alpha diversity and a lower proportion of butyrate-producing bacteria compared to the other cluster.

Within our unbiased clustering, we found a significant difference in the proportion of patients that were admitted to the ICU. The microbiome of patients admitted to the ICU can undergo marked compositional and functional changes, including a reduced alpha diversity and reduction of commensal bacteria [12]. These changes coincide with our findings in the second cluster. However, our sensitivity analysis showed that the long COVID differences persisted or were even amplified when we only included patients admitted to the ICU. As the patients were seen with a median of 176 days after infection, we would also expect to see (partial) recovery of their microbiome from antibiotics they might have received during their admission. This could hint at an altered rate of microbiome recovery after the antibiotic intake between our clusters, even among patients admitted to the ICU. Factors that could influence this recovery include diet and community factors [13]. Nine patients received antibiotics after their acute infection, including a single patient from cluster 2; however, none of these patients were admitted to the ICU.

There have been several studies exploring the microbiome of long COVID patients already. Research by Lui et al. [7] found that the microbiome in patients with COVID-19 who did not develop long COVID recovered to a composition similar to that of patients who did not experience COVID-19, while patients with long COVID showed increased abundances of *Ruminococcus gnavus* and a lower abundance of *Faecalibacterium prausnitzii*, coinciding with our clustering. Other clustering efforts also discovered a cluster with increased abundances of the *Clostridium* genus and *Ruminococcus gnavus* to be associated with a higher prevalence of respiratory complaints, similar to our cluster 2 [14]. This provides potential targets for targeted interventions. *Veillonella*, also showing increased abundances in cluster 2, has previously been linked to a lower DLCO at 3 months after infection [11]. However, *Veillonella* was not very common in our cohort, with the three species that were differentially abundant between the clusters present in only 7 to 10 patients.

We also observed a disparity in the proportion of butyrate-producing bacteria within our clusters. Butyrate is a SCFA and has been shown to reduce inflammation and gastrointestinal inflammation through the inhibition of NF-κB and histone acetylation [15]. Higher levels of butyrate in early life have also shown to reduce the chance to develop asthma in children [16], while the proportion of butyrate-producing bacteria reduced the risk of hospitalization after infection, including pulmonary infections [17]. Additionally, butyrate-producing bacteria were negatively correlated with the chance to develop long COVID [7]. Little is yet known about the relationship between butyrate and lung function. However, higher inflammation has been previously associated with a decreased FVC and DLCO in long COVID patients [18]. As butyrate reduces inflammation, this might constitute a protective role of these taxa for the lung function and partially explain the reduced lung function in the second cluster. Synbiotic interventions (combination of pre- and pro-biotics) targeting these butyrate-producing bacteria are freely available on the market and have already been applied to long COVID as well. In a Chinese cohort, supplementation with a synbiotic intervention significantly alleviated symptoms of fatigue, memory loss and general unwellness compared to placebo [19].

The main limitation of our study is our relatively small sample size and the resulting small size of cluster 2. This limits our power to detect clinically relevant differences. As a result, we were only able to confidently discern variables with large differences between our clusters, such as lung function and acute COVID-19 severity. However, our supervised analyses based on the alpha and beta diversity does reinforce that these main differences do exist within the microbiome of long COVID patients. Another limitation is that we did not have any microbiome samples from these patients before or at hospital admission. This means we do not know if the differences in their microbiome were already pre-existing, could have led to the differences in their long COVID or were simply a manifestation of their disease. Additionally, a control group of SARS-CoV-2 infected patients who did not develop long COVID could also provide valuable insights. However, our study is strengthened by the scope of the information we collected from our cohort. To our knowledge, we are the first to relate the gut microbiome of long COVID patients with lung function and CT-imaging and are able to relate the microbiome to other layers of information as well.

In this study, we found an association between the gastrointestinal microbiome, lung function and acute COVID-19. We found this association to be potentially mediated by a specific group of taxa and butyrate-producing bacteria. Targeting these specific taxa with a synbiotic or other interventions could pave the way for the development of a personalized approach for the treatment or monitoring of long COVID.

## 4. Materials and Methods

### 4.1. Patient Inclusion

Patients were enrolled as part of the Precision Medicine for More Oxygen (P4O2) COVID-19 study [4]. The P4O2 COVID-19 study is a multi-center observational study with the objective to identify long COVID patients that are at risk of developing chronic lung damage and to discover therapeutic biomarkers or to optimize lifestyle interventions for the treatment and prevention of long COVID. The inclusion criteria for the current analysis were (1) confirmed SARS-CoV-2 infection, (2) aged 40–65 years, (3) had an appointment at the post-COVID clinic at 3–6 months after infection, (4) had some understanding of the Dutch language, (5) had the ability to provide informed consent and (6) provided a fecal sample. Patients who were terminally ill or were involved in studies with marketed or investigational products within 4 weeks of inclusion were excluded from this study. This study was approved by the ethical committee of the Amsterdam University Medical Center under the identifier NL74701.018.20.

### 4.2. Study Visits and Patient Characteristics

Patients were seen across five different hospitals in the Netherlands at 3 to 6 months after the SARS-CoV-2 infection. This study visit is described in detail by Baalbaki et al. [4]. In short, the study visit included completing questionnaires, performing pulmonary function testing, CT-imaging of the thorax and biological sample collection. Patients also granted access to their medical records to collect patient characteristics and information regarding their acute COVID-19.

The administered questionnaires described the symptoms and impact of long COVID. Facets of their disease questioned were fatigue (Fatigue Severity Scale, FSS) [20], physical, mental and social wellbeing (Patient-Reported Outcomes Measurement Information System, PROMIS) [21], quality of life (EuroQol-5D, EQ-5D) [22], cognitive checklist after ICU admission (also administered if not admitted to the ICU, CLC-IC) [23], participation (Utrecht Scale for Evaluation of Rehabilitation–Participation, USER-P) [24] and anxiety and depression (Hospital Anxiety and Depression Scale, HADS) [25]. Pulmonary function was assessed with spirometry to measure the forced expiratory volume in one second (FEV_1_), forced vital capacity (FVC) and its ratio (FEV1/FVC). The diffusion capacity of the lung for carbon monoxide (DLCO) was assessed by body plethysmography. These measures were considered abnormal if they fell below the lower limit of normal, defined as the fifth percentile of a healthy population, adjusted for age, height and sex. For the acute phase of their acute COVID-19, the severity of their disease was classified according to WHO guidelines [26].

### 4.3. Sample Collection and DNA Isolation

Patients were asked to collect a fecal sample prior to the study visit. Total DNA was extracted using a repeated bead beating protocol. Two hundred and fifty mg of feces was homogenized in 700 µL of a S.T.A.R. buffer (Roche, Basel, Switzerland) by bead beating (FastPrep-24TM, MP Biomedicals, Santa Ana, CA, USA), and further lysed at 95 °C. The DNA was then cleaned with the Maxwell^®^ RSC Blood DNA Kit (Promega, Madison, WI, USA). Library preparation was performed with the NEBNext Ultra II FS DNA library prep kit with the NEBNext Multiplex Oligos for Illumina (New England Biolabs, Ipswich, MA, USA) according to the manufacturer’s instructions. The quality of the DNA was confirmed with the dsDNA HS kit (Thermo Fisher Scientific, Waltham, MA, USA) and Agilent High Sensitivity D5000 ScreenTape system (Agilent Technologies, Santa Clara, CA, USA). Sequencing was performed using the Illumina NovaSeq 6000 platform (Illumina, San Diego, CA, USA) to generate 2 × 150 bp paired-end reads.

### 4.4. Bacterial Read Processing

Bacterial read processing was performed according to the following pipeline. First, the quality of the reads was assessed with FastQC (v0.12.1) [27]. Next, Kneaddata (v0.12.0, https://github.com/biobakery/kneaddata, accessed on 22 July 2024), a wrapper around the packages Trimmomatic (v0.39) [28] and Bowtie2 (v2.2.3) [29] was used to filter out low quality reads and separate host and bacterial DNA. To ensure adequate separation of host and bacterial reads, reads were mapped to different versions of the human genome, GRCh38 and the newer CHM13v2.0 complete genome [30]. Finally, a bacterial taxonomy annotation and quantification was performed with MetaPhlAn (v4.0.6) using the ChocoPhlAnSGB database [31] with default parameters (ignoring archaea and eukaryotes).

### 4.5. Statistical Analysis

All analyses were performed in R (v4.1.2) with RStudio (v2021.09.1+372) [32]. Microbiome data were imported and handled using the phyloseq package (v1.38.0) [33]. Compositional diversity was compared between the samples to find interactions with clinical variables with the richness, Shannon diversity and the beta diversity based on the Bray–Curtis distance. Differential abundance between individual taxa was assessed with ANCOM-BC (v1.4.0) [34] with a Benjamini–Hochberg correction for multiple testing. Similarly to Kullberg et al. [17], we determined the proportion of the butyrate-producing bacteria in our samples. This was achieved by summing the counts of bacteria from known butyrate-producing genera (Appendix A) and dividing this by the total number of reads per sample.

For unsupervised clustering, taxon counts were first normalized using cumulative sum scaling with a log transformation. Secondly, pairwise distances between each patient were calculated with the Bray–Curtis distance. Then, patients were clustered using the ‘*hclust*’ function with the Ward.D2 method, incorporated into base R. The number of clusters was determined using the visual inspection of the cluster dendrogram, the silhouette index and bootstrapping stability. Finally, the clusters were compared for associations with clinical parameters, microbiome composition and proportion of butyrate-producing bacteria. As a sensitivity analysis, only patients admitted to the intensive care unit (ICU) were compared as well. Numerical variables were compared with a Student’s *t*-test or Wilcoxon signed-rank test, depending on the normality of the distribution. Categorical variables were compared with a Fisher’s exact test. An (adjusted) *p*-value below 0.05 was considered statistically significant.

## Figures and Tables

**Figure 1 ijms-26-01781-f001:**
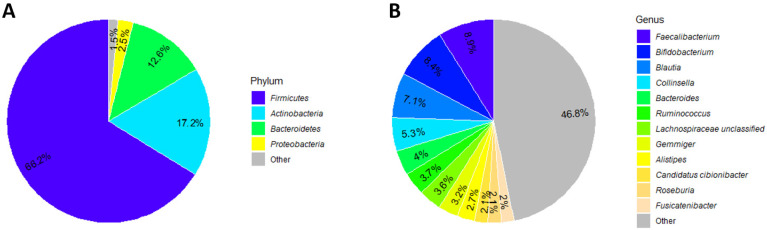
Cohort microbiome composition: pie charts illustrating the composition of the fecal microbiome of patients with long COVID at a phylum level (**A**) and at a genus level (**B**).

**Figure 2 ijms-26-01781-f002:**
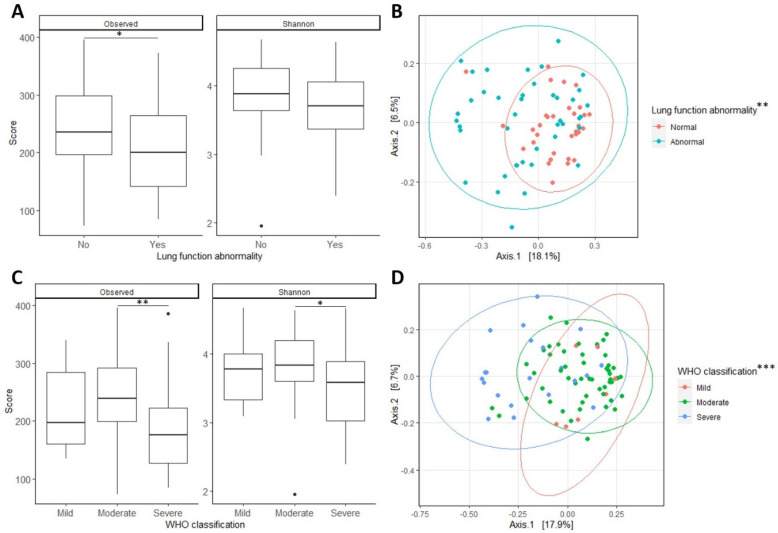
Cohort microbiome diversity: plots showing the alpha (**A**,**C**) and beta diversity (**B**,**D**) for the variables describing lung function abnormalities (**A**,**B**) and the WHO classification of their acute COVID-19 (**C**,**D**). * *p* < 0.05. ** *p* < 0.01. *** *p* < 0.001.

**Figure 3 ijms-26-01781-f003:**
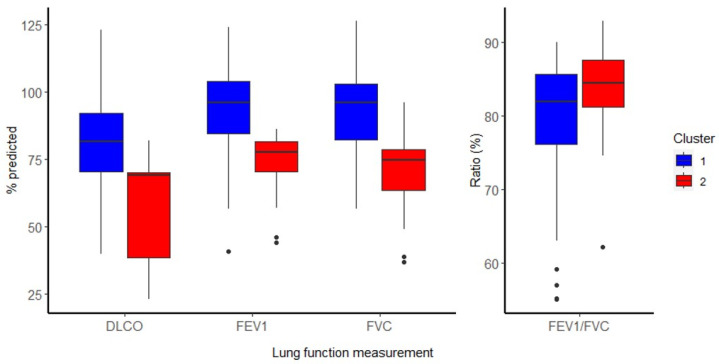
Cluster lung function: overview of the lung function measurements after cluster separation.

**Figure 4 ijms-26-01781-f004:**
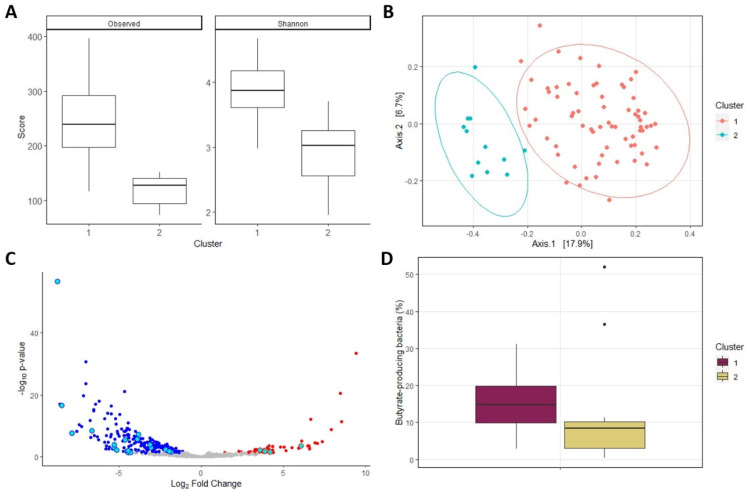
Cluster microbiome composition: plots depicting the differences in fecal microbiome composition between the long COVID clusters. (**A**): Alpha diversity. (**B**): Beta diversity. (**C**): Volcano plot of the taxa abundances. Blue taxa depict taxa more abundant in cluster 1, while red taxa are more abundant in cluster 2. Differentially abundant taxa that are part of butyrate-producing genera are highlighted with light blue. (**D**): Proportion of total counts that are part of butyrate-producing genera.

**Table 1 ijms-26-01781-t001:** Patient characteristics of all patients in the P4O2 COVID-19 study that were able to provide a valid fecal sample. Numerical data are shown as mean ± sd or median (IQR) while categorical data are shown as n/N (%).

	All Patients (N = 79)
**General Characteristics**	
Sex (Female)	42/79 (53.2%)
Age (Years)	54.6 ± 6.0
BMI (kg/m^2^)	30.4 ± 5.2
Time since SARS-CoV-2 infection (days)	176.0 (139.0, 197.0)
**Co-morbidities**	
Asthma	14/78 (17.9%)
COPD	5/78 (6.4%)
CVD	21/77 (27.3%)
Diabetes	11/78 (14.1%)
**Symptom Categories**	
Fatigue	57/79 (72.2%)
Respiratory	66/79 (83.5%)
Neurological	55/79 (69.6%)
Cardiovascular	21/79 (26.6%)
Gastrointestinal	23/79 (29.1%)
Other	17/79 (21.5%)
**Lung Function**	
FEV1 (% predicted)	n = 78; 90.6 ± 16.8
FVC (% predicted)	n = 78; 88.6 ± 18.0
FEV1/FVC (%)	n = 78; 80.4 ± 8.4
DLCO (% predicted)	n = 77; 78.1 ± 19.5
**Questionnaires**	
FSS	n = 71; 5.6 (4.0, 6.4)
PROMIS	n = 71; 28.6 ± 7.7
EQ-5D	n = 72; 9.0 (6.0, 11.0)
CLC-IC	n = 68; 5.0 (2.0, 8.0)
USER-P	n = 71; 80.0 (61.6, 95.3)
HADS Depression	n = 68; 4.0 (1.0, 7.2)
HADS Anxiety	n = 68; 3.0 (1.0, 7.0)
**CT Abnormalities**	
Ground-glass opacity/Consolidations	47/74 (63.5%)
Bronchiectasis	17/74 (23.0%)
Subpleural reticulation	20/74 (27.0%)
Lymphadenopathy	9/74 (12.2%)
Air trapping	10/74 (13.5%)
**Acute Phase Severity**	
Mild	7/79 (8.9%)
Moderate	51/79 (64.6%)
Severe	21/79 (26.6%)
**Acute Phase Complications**	
Hospital duration (Days)	n = 71; 8.0 (5.0, 16.0)
ICU admission	26/79 (32.9%)
Pulmonary embolism	14/77 (18.2%)
Thrombosis	13/76 (17.1%)
**Dominant SARS-CoV-2 Virus Type**	
Alpha	37/79 (46.8%)
Delta	33/79 (41.8%)
Omicron	9/79 (11.4%)

**Table 2 ijms-26-01781-t002:** Patient characteristics according to cluster. Numerical data are shown as mean ± sd or median (IQR) while categorical data are expressed as n/N (%). *p*-values are calculated using either a Wilcoxon signed-rank test or a Student’s *t*-test for numerical data, or a Fisher’s exact test for categorical data.

	Cluster 1 (N = 67)	Cluster 2 (N = 12)	*p*-Value
**General Characteristics**			
Sex (Female)	39/67 (58.2%)	3/12 (25.0%)	0.057
Age (Years)	54.5 ± 6.0	55.5 ± 6.1	0.591
BMI (kg/m^2^)	30.5 ± 5.0	29.9 ± 6.8	0.724
Time since SARS-CoV-2 infection (days)	176.0 (139.0, 192.5)	177.0 (154.0, 213.5)	0.386
**Co-morbidities**			
Asthma	11/66 (16.7%)	3/12 (25.0%)	0.443
COPD	5/66 (7.6%)	0/12 (0.0%)	1.000
CVD	17/65 (26.2%)	4/12 (33.3%)	0.726
Diabetes	8/66 (12.1%)	3/12 (25.0%)	0.360
**Symptom Categories**			
Fatigue	50/67 (74.6%)	7/12 (58.3%)	0.299
Respiratory	54/67 (80.6%)	12/12 (100.0%)	0.199
Neurological	47/67 (70.1%)	8/12 (66.7%)	1.000
Cardiovascular	20/67 (29.9%)	1/12 (8.3%)	0.166
Gastrointestinal	21/67 (31.3%)	2/12 (16.7%)	0.429
Other	13/67 (19.4%)	4/12 (33.3%)	0.276
**Lung Function**			
FEV1 (% predicted)	n = 66; 93.8 ± 15.1	72.5 ± 14.8	**<0.001**
FVC (% predicted)	n = 66; 92.1 ± 15.7	69.2 ± 18.3	**<0.001**
FEV1/FVC (%)	n = 66; 79.9 ± 8.4	82.9 ± 8.2	0.256
DLCO (% predicted)	n = 66; 81.7 ± 16.9	n = 11; 56.7 ± 20.7	**<0.001**
**Questionnaires ***			
FSS (↓)	n = 62; 5.5 (3.9, 6.4)	n = 9; 5.6 (5.1, 6.2)	0.710
PROMIS (↑)	n = 62; 29.1 ± 7.7	n = 9; 25.6 ± 7.6	0.202
PC-PTSD-5 (↓)	n = 61; 1.0 (0.0, 2.0)	n = 10; 1.0 (0.0, 1.0)	0.902
EQ5D (↓)	n = 62; 9.0 (6.0, 11.0)	n = 10; 11.5 (7.8, 12.8)	0.084
CLCIC (↓)	n = 58; 4.5 (2.0, 8.0)	n = 10; 5.5 (3.2, 7.8)	0.821
USER-P (↑)	n = 62; 83.3 (66.7, 96.7)	n = 9; 62.5 (53.3, 80.0)	0.114
HADS Depression (↓)	n = 59; 4.0 (1.0, 6.5)	n = 9; 7.0 (2.0, 8.0)	0.287
HADS Anxiety (↓)	n = 60; 3.5 (1.8, 7.2)	n = 8; 1.0 (0.0, 3.8)	0.092
**CT Abnormalities**			
Ground-glass opacity/Consolidations	39/62 (62.9%)	8/12 (66.7%)	1.000
Bronchiectasis	10/62 (16.1%)	7/12 (58.3%)	**0.004**
Subpleural reticulation	17/62 (27.4%)	3/12 (25.0%)	1.000
Lymphadenopathy	8/62 (12.9%)	1/12 (8.3%)	1.000
Air trapping	9/62 (14.5%)	1/12 (8.3%)	1.000
**Acute Phase Severity**			
Mild	7/67 (10.4%)	0/12 (0.0%)	**<0.001**
Moderate	48/67 (71.6%)	3/12 (25.0%)	
Severe	12/67 (17.9%)	9/12 (75.0%)	
**Acute Phase Complications**			
Hospital duration (Days)	n = 59; 7.0 (4.5, 12.0)	47.5 (15.8, 58.2)	**<0.001**
ICU admission	15/67 (22.4%)	11/12 (91.7%)	**<0.001**
Pulmonary embolism	9/65 (13.8%)	5/12 (41.7%)	**0.036**
Thrombosis	7/65 (10.8%)	6/11 (54.5%)	**0.002**
Antibiotic use	17/67 (25.4%)	9/12 (75.0%)	**0.002**
**Dominant SARS-CoV-2 Virus Type**			
Alpha	31/67 (46.3%)	6/12 (50.0%)	1.000
Delta	28/67 (41.8%)	5/12 (41.7%)	
Omicron	8/67 (11.9%)	1/12 (8.3%)	

* Arrows indicate if a higher or lower score is more desirable for the patient.

## Data Availability

The fecal microbiome data generated for this study are publicly available on BioProject under the BioProject ID: PRJNA1181655 (reviewer link). The clinical data are not publicly available due to agreements made by the consortium that only allow access to the data by consortium partners that answer pre-specified research questions. A request for access to data by organizations outside of the consortium can be submitted to the P4O2 Data Committee (via p4o2@amsterdamumc.nl).

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
