# Peer review of "Exploring Heterogeneity of Fecal Microbiome in Long COVID Patients at 3 to 6 Months After Infection"

_ijms, 2025, doi:10.3390/ijms26041781_

Round 1

Reviewer 1 Report

Comments and Suggestions for Authors

Abstract:

Maybe it would be better to specify what cluster 1 and 2 means. Please, write the taxonomy of the bacteria correctly.

Introduction:

Lines 42 – 44: Please revise.

Lines 55 – 56: How culture influences the microbiome?

Lines 58: Veillonella spp.?  

Please, use the past tense for the objectives of the study.

Materials and Methods:

Section 2.4. Read processing is not very clear to me.

Line 133-134: Similarly to Kull-133 berg et al., [27] we determined the proportion of the butyrate producing bacteria in our 134 samples. Why? If that is relevant, paraphs the authors should consider to arguments it more, even in the introduction.

Is there a control group?

Results:

Please, be careful with the scientific classification of organisms. Use italics for bacteria phyla.

Fig 1. Revise the taxonomy and improve the figure resolution. At which month of collection (3 or 6) corresponds to this microbial composition?

Fig 2. Why do the authors use different p values?

Lines 184-187, please, elaborate more. Which is the difference between these two clusters in terms of bacterial composition? Figure S2. I cannot read the samples, and I can see various clusters. Please, indicate those clusters in the figure.

What about the microbial composition in these patients at 3 to 6 months after infection? In which month were the samples collected and analyzed?

Line 220: A full list of differentially abundant taxa can be found in the supplementary files. Why are things highlighted in red or blue?

Discussion:

Lines 239-240: We found that patients with at least one abnormality in the lung function test measures or a severe acute COVID-19 showed a decreased alpha diversity and a separation for the beta diversity compared to the other patients.

Patients compared with other patients. Please, be more specific. Were those patients compared against a control group?

See space between words.

Conclusion:

The conclusion section is missing.

Other specific points:

* These authors have contributed equally – I can see only one.

Remove highlighter green

Missing info: Institutional Review Board Statement and Informed Consent Statement

Reviewer 2 Report

Comments and Suggestions for Authors

I agree with the limitations of the study and its design.

Although you showed changes in the gut microbiome characterized by lacking diversity and reduced flora that produces butyrate, I do not see data from a control group (ICU patients with similar SOFA scores, SARS CoV2 negative, or another parameter) that allows for comparison. These patients were already sick with multiple co-morbid conditions; the use of antibiotics is common practice in acutely ill patients in the ICU, and those are some situations that will affect a short and long-term microbiome. I wonder if those changes exclude the post-ICU Long COVID vs. post-ICU no SARS CoV2 population.

Patients with severe acute SARS CoV2 have more iatrogenic interventions that lead to more complications (ETT, administration of high O2, antibiotics, immunomodulatory drugs, etc) that have nothing to do with changes in the microbiome. 

A more homogenous population, with baseline and follow-up microbiome analysis, will help clarify the impact of SARS-CoV on Long COVID.

Do you have any comments on respiratory microbiome as a better approach than fecal?

Author Response

I agree with the limitations of the study and its design.

Although you showed changes in the gut microbiome characterized by lacking diversity and reduced flora that produces butyrate, I do not see data from a control group (ICU patients with similar SOFA scores, SARS CoV2 negative, or another parameter) that allows for comparison. These patients were already sick with multiple co-morbid conditions; the use of antibiotics is common practice in acutely ill patients in the ICU, and those are some situations that will affect a short and long-term microbiome. I wonder if those changes exclude the post-ICU Long COVID vs. post-ICU no SARS CoV2 population.

Unfortunately we did not have a control group with similar data to compare against, and only explored the differences between long COVID patients. Adding this control group to our analysis would be valuable, especially patients that were admitted to the hospital with SARS-CoV-2, but did  not develop long COVID, in order to distinguish potential mechanisms leading to long COVID. However, our patients were recruited from the long COVID outpatient clinics in the participating hospitals. This means that the only people we were able to recruit were long COVID patients, and only after at least three months of having persistent symptoms after confirmed SARS-CoV-2 infections.

Patients with severe acute SARS CoV2 have more iatrogenic interventions that lead to more complications (ETT, administration of high O2, antibiotics, immunomodulatory drugs, etc) that have nothing to do with changes in the microbiome. 

We definitely agree with this point, and this was one of our main concerns as well. To clarify if this was the main source of differences between the clusters we performed a sensitivity analysis where we only compared the composition and clinical characteristics of patients that were admitted to the ICU during their visit. This was described in the manuscript, although briefly:

“When examining only those patients admitted to the ICU, we found that most of the conclusions regarding the entire cohort also apply to this subset of patients, or are even more significant (Table S2).” (lines 204-206)

“When only studying the patients admitted to the ICU, we still observed a significantly lower proportion of butyrate producing bacteria in cluster 2 (p = 0.047, Figure S3).” (lines 230-232)

More information can also be found in Table S2 (comparison of clinical characteristics) and Figure S3 (comparison of butyrate producing bacteria between only these patients).

A more homogenous population, with baseline and follow-up microbiome analysis, will help clarify the impact of SARS-CoV on Long COVID.

We also agree with this point, and this is a limitation of our study that we were unable to resolve, so we have mentioned this in the limitation section of the discussion.

“Another limitation is that we did not have any microbiome samples from these patients before or at hospital admission. This means we do not know if the differences in their microbiome were already pre-existing, could have led to the differences in their long COVID, or simply a manifestation of their disease.” (lines 295-299)

Do you have any comments on respiratory microbiome as a better approach than fecal?

Long COVID is a multi-system disorder, not strictly respiratory (PMID: 36639608). As a result we decided to go for a more systemic approach in the gastrointestinal microbiome, as this has shown to have multi-systemic consequences through, for example, the gut-lung axis and gut-brain axis (PMID: 30976087, PMID: 36232548). It is unlikely that the gastrointestinal microbiome explains everything that can develop into long COVID, other molecular layers such as the transcriptome or epigenome could also provide explanations. In that sense we would not call the respiratory microbiome a better or worse approach to explore long-COVID, but it could provide additional insights into the disease pathology that is not explained by other sources of information, just as the gastrointestinal microbiome might provide that information.
